# Cardiac Tissue-like 3D Microenvironment Enhances Route towards Human Fibroblast Direct Reprogramming into Induced Cardiomyocytes by microRNAs

**DOI:** 10.3390/cells11050800

**Published:** 2022-02-25

**Authors:** Camilla Paoletti, Elena Marcello, Maria Luna Melis, Carla Divieto, Daria Nurzynska, Valeria Chiono

**Affiliations:** 1Department of Mechanical and Aerospace Engineering, Politecnico di Torino, 10129 Turin, Italy; elena.marcello@polito.it (E.M.); lunamelis92@gmail.com (M.L.M.); valeria.chiono@polito.it (V.C.); 2Centro 3R (Interuniversity Center for the Promotion of 3Rs Principles in Teaching and Research), Lucio Lazzarino 1, 56122 Pisa, Italy; 3Istituto Nazionale di Ricerca Metrologica, Division of Advanced Materials and Life Sciences, 10135 Turin, Italy; c.divieto@inrim.it; 4Department of Medicine, Surgery and Dentistry “Scuola Medica Salernitana”, 84084 Salerno, Italy; dnurzynska@unisa.it

**Keywords:** direct reprogramming, human fibroblast, induced cardiomyocytes, microRNAs, miRcombo, cardiac extracellular matrix, three-dimensional culture, hydrogels

## Abstract

The restoration of cardiac functionality after myocardial infarction represents a major clinical challenge. Recently, we found that transient transfection with microRNA combination (miRcombo: miR-1, miR-133, miR-208 and 499) is able to trigger direct reprogramming of adult human cardiac fibroblasts (AHCFs) into induced cardiomyocytes (iCMs) in vitro. However, achieving efficient direct reprogramming still remains a challenge. The aim of this study was to investigate the influence of cardiac tissue-like biochemical and biophysical stimuli on direct reprogramming efficiency. Biomatrix (BM), a cardiac-like extracellular matrix (ECM), was produced by in vitro culture of AHCFs for 21 days, followed by decellularization. In a set of experiments, AHCFs were transfected with miRcombo and then cultured for 2 weeks on the surface of uncoated and BM-coated polystyrene (PS) dishes and fibrin hydrogels (2D hydrogel) or embedded into 3D fibrin hydrogels (3D hydrogel). Cell culturing on BM-coated PS dishes and in 3D hydrogels significantly improved direct reprogramming outcomes. Biochemical and biophysical cues were then combined in 3D fibrin hydrogels containing BM (3D BM hydrogel), resulting in a synergistic effect, triggering increased CM gene and cardiac troponin T expression in miRcombo-transfected AHCFs. Hence, biomimetic 3D culture environments may improve direct reprogramming of miRcombo-transfected AHCFs into iCMs, deserving further study.

## 1. Introduction

Ischemic heart disease is the main cause of death worldwide [1]. During myocardial infarction (MI), billions of cardiomyocytes are irreversibly lost, leading to scar tissue formation, followed by left ventricle remodeling, which may result in progressive heart failure. Over the past few decades, several therapeutic approaches have been investigated with the aim to recover cardiac functionality, encompassing tissue engineering strategies as well as cell-based and gene therapies [2,3]. Among them, direct reprogramming of fibroblasts into induced cardiomyocytes (iCMs) has emerged as a promising approach with the potentiality to reverse post-MI cardiac fibrosis, restoring cardiomyocyte population. Since the first study in 2010 reporting the effective transdifferentiation of mouse fibroblasts into iCMs through the expression of Gata4, Mef2c and Tbx5 cardiac transcription factors (TFs), direct reprogramming has expanded rapidly in the scientific community [4,5]. Over the years, different direct reprogramming strategies have been investigated, including the expression of cardiac lineage specific TFs by viral vectors [4], the use of small molecules [6], the suppression of fibroblast signatures [7], the expression of cardiac microRNAs (miRs-1, 133, 208 and 499, called “miRcombo”) [8], or combinations of these methods [9]. In particular, we demonstrated that a single transient transfection with miRcombo, previously validated in vitro [8] and in vivo [10] in a mouse model, was able to trigger direct reprogramming of adult human cardiac fibroblasts into iCMs, as demonstrated by the expression of cardiac lineage specific TFs and cardiomyocyte markers and spontaneous calcium flux in reprogrammed cells [11]. However, direct reprogramming of human adult cardiac fibroblasts into iCMs showed limited efficiency, with ~11%.cardiac troponin T (cTnT)-positive cells after 15 days post-transfection [11].

In vitro culture of transfected fibroblasts onto cardiac tissue-like substrates could help in overcoming barriers to human cell reprogramming, improving its efficiency [12]. One relevant example of a culture microenvironment able to provide biomimetic biophysical and biochemical cues is represented by three-dimensional (3D) hydrogels with biomimetic stiffness based on cardiac extracellular matrix (ECM) proteins [13]. Indeed, murine fibroblasts cultured on the surface of Matrigel-coated poly(acrylamide) hydrogels with biomimetic stiffness (8 kPa) and transfected with Gata4, Mef2c and Tbx5 and Hand2 (GMTH) were reprogrammed into iCMs more efficiently compared to cells cultured onto stiff polystyrene (PS) tissue culture plates [14]. Moreover, the efficiency of miRcombo-mediated direct reprogramming of mouse fibroblasts into iCMs was also enhanced by an in vitro culture of transfected fibroblasts in 3D fibrin/Matrigel hydrogels rather than on 2D PS plates [15].

However, studies on the effect of the culture microenvironment on direct reprogramming are limited and the majority of them employed Matrigel, an extract from the basement membrane of Engelbreth–Holm–Swarm (EHS) mouse tumors. Matrigel is able to support cell culture due to its composition, mainly consisting of laminin (~60%), collagen IV (~30%), entactin (~8%) and the heparin sulfate proteoglycan perlecan (~2–3%) [16]. However, due to its animal origin, Matrigel contains xenogenic contaminants. As an alternative to Matrigel, a culture microenvironment based on cardiac ECM proteins could be exploited to improve direct reprogramming efficiency. Cardiac ECM is a complex 3D network of fibrillar proteins, such as fibrillar collagens type I and III, and non-fibrillar proteins, such as fibronectin, laminin and collagen IV, which constitute the basement membrane together with hyaluronic acid (HA), proteoglycans and glycosaminoglycans (GAGs) [17]. Cardiac fibroblasts are the main responsible cells for the synthesis and remodeling of ECM proteins which build up a myocardial tissue architecture in both physiological and pathological conditions [17]. During embryonic and fetal heart development, cardiac ECM proteins support myocardial cell precursor migration and polarization, tightly regulating cardiac tissue development and organization [17]. Interestingly, culture substrates containing cardiac ECM proteins could also improve the maturation of human induced-pluripotent stem cell-derived cardiomyocytes (hiPSC-CMs) [18]. The addition of cardiac ECM protein microparticles derived from human cardiac tissue into 3D hiPSC-CM aggregates improved their sarcomeric organization and calcium handling [18]. Recently, Sauls et al. found that collagens and laminins are upregulated in mouse embryonic fibroblasts (MEFs) at 48 and 72 h post-transduction with Mef2c, Gata4, Tbx5 cardiac TFs. This result demonstrates that, in the early stages of fibroblast reprogramming, cells recreate their native microenvironment, which in turn improves their transdifferentiation into iCMs [19]. This finding is in agreement with previous reports on the beneficial role of poly(ethylene glycol) (PEG) hydrogels functionalized with arginine-glycine-aspartic acid (RGD) and laminin in enhancing the direct reprogramming efficiency of murine fibroblasts into iCMs [20]. Indeed, high concentrations of laminin and RGD peptide increased the formation of α-sarcomeric actinin-positive and beating iCMs from reprogrammed MEFs [20].

However, to the best of our knowledge, studies on the role of the culture microenvironment on the direct reprogramming efficiency of human fibroblasts into iCMs have not been carried out to date [14,15,20]. Herein, we investigated the effect of cardiac ECM proteins on the in vitro miRcombo-mediated direct reprogramming of human adult cardiac fibroblasts into iCMs. Initially, an in vitro cardiac ECM called biomatrix (BM) was produced by means of a long-term in vitro culture of human adult cardiac fibroblasts followed by decellularization. Then, it was exploited for in vitro cell culture, providing biochemical cues favoring direct reprogramming [21]. The culturing of miRcombo-transfected cells on BM-coated PS tissue culture plates enhanced direct reprogramming efficiency, as demonstrated by droplet digital PCR (ddPCR) analysis and flow cytometry. Then, the effect of 3D fibrin hydrogel (3D hydrogel) on the direct reprogramming efficiency of human adult cardiac fibroblasts into iCMs was also studied by culturing miRcombo-transfected cells on the surface or within fibrin hydrogels, compared to cell cultures on PS plates. The 3D culture in fibrin hydrogel significantly improved direct reprogramming efficiency with respect to the other tested culture conditions, as assessed by ddPCR analysis. Finally, miRcombo-transfected cells were cultured in 3D fibrin hydrogels containing BM (3D BM hydrogel), providing cardiac tissue-mimetic biophysical and biochemical cues to reprogramming cells. The expressions of cardiomyocyte genes and cTnT were significantly enhanced in cells cultured for 15 days within 3D BM compared to 3D hydrogels. Overall, the results demonstrate that a biomimetic 3D culture microenvironment can enhance the direct reprogramming efficiency of miRcombo-transfected human adult cardiac fibroblasts into iCMs. Future investigations will elucidate which molecular barriers to direct reprogramming can be overcome by the use of 3D biomimetic cell culture substrates, paving the way for the research into more efficient strategies for direct cardiac reprogramming [13].

## 2. Materials and Methods

### 2.1. Primary Cell Culture

Normal human atrial cardiac fibroblasts (AHCFs) were purchased from Lonza (CC-2903) (Walkersville, MD, USA) and were maintained in culture using Fibroblasts Growth Medium-3 (FGM-3, Lonza, CC-4526). Cells were expanded until passage four and then used for experiments.

### 2.2. Biomatrix (BM) Production In Vitro

AHCFs (5 × 10^5^ cells/dish) were cultured in 100 mm-diameter Petri dishes with FGM-3 medium for up to 21 days. The decellularization process was performed following a previously described protocol [21]. Briefly, cells were removed by incubation with a solution of 0.25% Triton X- 100 and 10 mM NH_4_OH in phosphate buffered saline (PBS, Gibco, Waltham, MA, USA) prewarmed to 37 °C. Petri dishes were washed twice with PBS and then frozen for 24 h and lyophilized using CoolSafe 4-15L freeze-dryers (Labogene, Scandinavia) for 24 h. Afterwards, lyophilized BM was collected from wells in the form of powder and stored at −80 °C.

### 2.3. BM Immunofluorescence

AHCFs (3 × 10^4^ cells/dish) were plated on 35 mm µ-Dishes (Ibidi, Gräfelfing, Germany) and cultured for 21 days as described before. To characterize cellularized samples before decellularization, cells were fixed in 4% paraformaldehyde (PFA, Alfa Aesar, Ward Hill, MA, USA), permeabilized with permeabilization buffer (0.01% Triton X 100) for 10 min and blocked with bovine serum albumin (BSA) 1% in PBS for 1 h. To characterize BM after decellularization, samples were decellularized as previously described, fixed in 4% PFA and blocked with BSA 1% in PBS for 1 h. Both sample types were then incubated with primary antibodies against vimentin (V2258 Sigma-Aldrich, St. Louis, MO, USA), discoidin domain receptor 2 (DDR2, MA5-15356 Invitrogen, Waltham, MA, USA), alpha-smooth muscle actin (α-SMA, A7607 Sigma-Aldrich), laminin (L827-1 Sigma-Aldrich), fibronectin (F3648 Sigma-Aldrich), collagen I (C2456 Sigma-Aldrich) and collagen IV (ab6586 Abcam, Cambridge, UK). After washing with PBS, samples were stained with Alexa-488 (120077 Invitrogen) and 555 secondary antibodies (A21422 Invitrogen). Nuclei were counterstained with 4′,6-diamidino-2-phenylindole (DAPI, Invitrogen) as previously described. Images were acquired using the Spinning Disk Ti-2 Eclipse microscope (Nikon, Tokyo, Japan) at 20× magnification and merged using ImageJ software (Fiji). For vimentin, DDR2 and α-SMA, five randomly chosen microscopic fields were taken for each sample (*n* = 3) and positive cells for the protein of interest were manually counted using ImageJ software and reported as the average number of positive cells/total nuclei. Experiments were performed in biological and technical triplicates.

### 2.4. BM Coating of Tissue Culture PS Plates

BM powder was added to PBS at 100 μg/mL concentration, in the presence of a warm water bath at 37 °C, until complete BM dissolution. Then, the solution was filtered using syringe filters with 0.22 µm pore size (Carlo Erba, Milan, Italy). The BM solution was used to coat the surface of 12-, 24- and 96-well plates (300, 200 and 100 µL of BM solution, respectively) and incubated for 2 h at 37 °C. Finally, the excess of the BM coating solution was removed, and the coated surfaces were washed with PBS twice prior to cell culture.

### 2.5. Cell Adhesion on BM-Coated Tissue Culture PS Plates

Tissue culture 96-well plates were coated with BM as previously described. AHCFs were plated on BM-coated dishes at a density of 3 × 10^3^ cells/well in Dulbecco’s modified Eagle medium (DMEM) high glucose (Gibco), 10% FBS (Sigma-Aldrich) and 1% glutamine (Sigma-Aldrich). Uncoated surfaces were used as the control. At 0.5, 1, 2 and 4 h post culture, each well was gently washed with fresh PBS, in order to remove floating cells, and then cells were fixed with 4% PFA for 10 min at room temperature. After PBS washing, DAPI was added to each well for 10 min in the dark at a final concentration of 1 μg/mL. Lastly, each well was rinsed two times with PBS. Fluorescence microscopy images of adherent cells were acquired using the Spinning Disk Ti-2 Elipse microscope (Nikon) at 10× magnification. All the conditions were performed in biological and technical triplicates and cell adhesion was quantified with ImageJ (Fiji) by determining the average number of attached cells from 5 randomly chosen microscopic fields for each sample.

### 2.6. Cell Transfection

Cell transfection was performed as previously described [11]. AHCFs were plated in 6-well plates (1.1 × 10^5^ cells) 24 h before transfection. Cells were transfected using DharmaFECT 1 (Dharmacon™) with miRcombo (miR-1-3p, miR-133a-3p, miR-208a-3p and miR-499a-5p, mirVana^®^ miRNA mimic, Life Technologies) or negmiR (Negative Control #1, mirVana™ miRNA Mimic, Life Technologies) according to manufacturer’s instructions, in DMEM high glucose, 10% FBS and 1% glutamine. After 24 h, transfection medium was removed and cells were cultured for different time intervals depending on the different experimental conditions described below in DMEM high glucose, 10% fetal bovine serum (FBS), 1% glutamine and 1% penicillin-streptomycin.

### 2.7. Cell Culture on BM-Coated Tissue Culture PS Plates

After miRcombo transfection, described in the previous paragraph, the cell culture was continued on the same well for control conditions on uncoated PS dishes or cells were trypsinized and then plated on BM-coated wells. Culturing was continued for up to 15 days.

### 2.8. Cell Culture Using Fibrin-Based Hydrogels

AHCF culturing using fibrin hydrogels was performed in both two-dimensional (2D) and three-dimensional (3D) conditions: (A) AHCFs were cultured on the surface of fibrin hydrogels (2D hydrogel); (B) AHCFs were embedded within 3D fibrin hydrogels (3D hydrogel); and (C) AHCFs were embedded within fibrin hydrogel containing BM (3D BM hydrogel). In parallel, control cell experiments on PS plates were also performed. In detail, fibrin hydrogels (A–C) were prepared using a final concentration of 5 mg/mL fibrinogen (Sigma-Aldrich), dissolved in warm PBS, and 50 unit/mL thrombin (Sigma-Aldrich) in 0.1% BSA (Sigma-Aldrich) in PBS. For the C condition only, BM solution was added to fibrinogen solution at a final concentration of 100 μg/mL before adding thrombin solution. Cells transfected as described in Section 2.6 were cultured in different conditions: (A) on the top of 2D hydrogels (by mixing 200 µL of fibrinogen 2X and 200 µL of thrombin 2× solutions), (B) added to fibrinogen solution (by mixing 20 µL of fibrinogen 2X and 20 µL of thrombin 2× solutions) and (C) added to fibrinogen and BM solution (by mixing 10 µL of fibrinogen 4×, 20 µL of BM 2× and 10 µL of thrombin 4× solutions) at a final concentration of 1 million cells/mL. Hydrogels were then placed at 37 °C for 45 min, and 150 µL of medium containing Dulbecco’s modified Eagle medium (DMEM) high glucose (Gibco) with 10% FBS (Sigma-Aldrich), 1% glutamine (Sigma-Aldrich) and 1 mg/mL aminocaproic acid (Sigma-Aldrich) was added. The culture was continued for up to 15 days. Three-dimensional hydrogels without cells for immunofluorescence analysis were formed as described above without embedding cells and immediately fixed.

### 2.9. RNA isolation and RT-PCR

After 15 days of culture, total RNA was extracted using QIAzol Lysis Reagent (Qiagen, Hilden, Germany) according to the manufacturer’s instructions. For 2D and 3D hydrogels, RNA was extracted with QIAzol using Tissue Lyser (Qiagen) for 1 min at medium frequency. RNA concentration and quality were assessed using NanoQuant plate (Tecan Group Ltd., Männedorf, Switzerland). cDNA was obtained using the High-Capacity cDNA Reverse Transcription Kit (Applied Biosystems, Waltham, MA, USA).

### 2.10. Gene Expression by Droplet Digital PCR

The expression of cardiac troponin T2 (TNNT2, ID assay: dHsaCPE5052344), sodium voltage-gated channel alpha subunit 5 (SCN5A, ID assay: dHsaCPE5042208), myosin light chain 7 (MYL7, ID assay: qHsaCEP0050426), calcium voltage-gated channel subunit alpha1 C (CACNAC1, ID assay: dHsaCPE5038360) and actin alpha cardiac muscle 1 (ACTC1, ID assay: qHsaCIP0028009) at 15 days of culturing was examined by droplet digital PCR (ddPCR) (Bio-Rad Laboratories, Hercules, CA, USA), allowing the quantification of miniscule amounts of template and able to discriminate very small differences in gene expression [22]. Primers and probes (Bio-rad Laboratories) labeled with carboxyfluorescein (FAM)/hexachlorofluorescein (HEX) were mixed with 10 µL of ddPCR Super-mix for probes (no dUTP), 5 µL containing 20 ng cDNA and 4 µL H_2_O up to a final volume of 20 µL/reaction. The 20 μL reaction mixture was then loaded into the Bio-Rad DG8 disposable droplet generator cartridge. A volume of 70 μL of droplet generation oil was loaded into the oil well for each sample. The cartridge was placed into the QX100 droplet generator (Bio-Rad). Droplet generation was performed according to manufacturer’s instructions. The generated droplets were transferred to a ddPCR™ 96-well PCR plate (Bio-Rad) and placed on a T100 thermal cycler (Bio-Rad). Thermal cycling conditions were 95 °C for 10 min (1 cycle), 94 °C for 30 s and 55 °C for 30 s (40 cycles), 98 °C for 10 min (1 cycle), and 4 °C infinite hold. Then, the PCR plate was loaded on a BioRad QX100 droplet reader and processed for DNA copy number quantification. Analysis of the ddPCR data was performed by QuantaSoft analysis software (Bio-Rad Laboratories). No template control was included in each assay. Experiments were performed in biological and technical triplicate. Glyceraldehyde 3-phosphate dehydrogenase (GAPDH; ID assay: dHsaC-PE5031597) was used as a housekeeping gene to perform quantitative normalization. Results were reported as the concentration (cDNA copies/µL) of the gene of interest normalized on the concentration mean (cDNA copies/µL) of GAPDH.

### 2.11. Flow Cytometry

For flow cytometry analysis, cells cultured for 15 days were treated with 0.05% Trypsin/EDTA (Sigma Aldrich) and permeabilized with 0.5% *v/v* Tween 20 in PBS for 5 min. Ice-cold PBS with 10% FBS and 1% sodium azide (Sigma Aldrich) was used for washing between each step. Cells were incubated with cardiac troponin T (cTnT) primary antibody (701620 Invitrogen) for 1 h at 4 °C and Alexa Fluor 488-conjugated secondary antibody (Abcam) for 1 h at 4 °C in the dark. Cells were analyzed using a Guava EasyCyte (Merck, Kenilworth, NJ, USA) flow cytometer, and data analysis was performed using GuavaSoft 3.2.

### 2.12. Immunofluorescence Analysis of Hydrogels

For 3D hydrogels, hydrogels were fixed in 4% paraformaldehyde for 20 min, embedded into FSC 22 OCT liquid (Leica, Wetzlar, Germany) and then sectioned using Leica CM1950 cryostat (30 µm). Samples were then permeabilized for 20 min and blocked as previously described for 1 h. Cells were incubated with primary antibodies against fibronectin, laminin, collagen IV and cardiac troponin T (Invitrogen) overnight at 4 °C. After washing with PBS, samples were stained with Alexa-488 secondary antibody (Invitrogen) for 2 h at room temperature. For cellularized hydrogels, nuclei were counterstained with DAPI (Invitrogen) as previously described. Images were acquired using Spinning Disk Ti-2 Elipse microscope (Nikon) at 20× and 60× magnification and merged using ImageJ software (Fiji). The percentage of cTnT-positive cells was calculated by counting the number of cTnT-positive cells with respect to the total number of cells, identified using nuclei.

### 2.13. Rheological Characterization of Hydrogels

Rheological characterization of fibrin-based hydrogels was performed using a MCR302 Anton Paar GmbH rheometer (Graz, Austria) coupled with a 25 mm parallel plate geometry. Samples were prepared as previously described in Section 2.8 with a final volume of 500 µL. All the analyses were carried out at 37 °C. Strain sweep tests were performed to identify the linear viscoelastic region (LVR) of the samples (strain varying from 0.1 to 100%, at an oscillatory frequency of 1 Hz, ω = 2πf = 6.28 rad/s). The viscoelastic properties of hydrogels (i.e., G’, modulus and G’’, loss modulus) were evaluated through oscillatory frequency sweep measurements (frequency varying from 0.1 to 10 rad/s) at a strain amplitude of 1%, accordingly to the LVR region. All rheological tests were conducted in triplicate.

### 2.14. Scanning Electron Microscopy (SEM) of Hydrogel Sections

Three-dimensional hydrogel and 3D BM hydrogel internal sections were examined by SEM. Hydrogel samples were formed as described above and fixed with 4% paraformaldehyde at room temperature, then frozen for 24 h and lastly lyophilized using CoolSafe 4-15L freeze-dryers, Labogene (Scandinavia) for 24 h. The samples were fractured with the aid of a bistoury in liquid nitrogen, and then sputter coated with gold. SEM images of the sections were acquired at different magnifications.

### 2.15. Calcium Transient Measurement of Hydrogels

Calcium transient was analyzed in miRcombo transfected cells cultured in 3D and 3D BM hydrogels for 15 days (following the procedure described in Section 2.8). To record calcium transients, cells were loaded with 5 μM of Fluo-4 AM (Invitrogen) in modified Tyrode’s solution (140 mM NaCl, 5 mM KCl, 1.8 mM CaCl_2_, 1 mM MgCl_2_, 10 mM glucose and 10 mM Hepes) with 0.1% bovine serum albumin (BSA) at 37 °C for 30 min while shielded from light. Cells were washed in modified Tyrode’s solution at 37 °C for 30 min. Calcium transient at 5 fps was recorded using Nikon Eclipse Ti2 spinning disk and NIS-Elements software (Nikon). Caffeine (Sigma-Aldrich) solution (20 mM) was added to each sample while peforming the time-lapse. At least 5 fields were observed for each sample. Time lapses were subsequently converted into frames, imported and analyzed using Image J software (NIH), normalizing the Ca^2+^ signal fluorescence (F) to background fluorescence (F_0_).

### 2.16. Statistical Analysis

The results are shown as means ± standard error of the mean (SEM) of triplicate experiments. Statistical analyses were performed with Student’s t-test, with *p*-value reported as * *p* < 0.05 considered statistically significant, ** *p* < 0.01 considered highly significant and *** *p* < 0.0001 very highly significant. All graphs were prepared using GraphPad software.

## 3. Results

### 3.1. BM Production by In Vitro Culture of Human Cardiac Fibroblasts

BM, an in vitro cardiac-like ECM, was prepared by plating and culturing AHCFs for 21 days on PS culture plates. Then, samples were decellularized and BM was collected in powder form, ready for solubilization (Figure 1a).

Initially, we investigated whether the long-term culturing of AHCFs on PS plates culture plates could induce their transition into a myofibroblast phenotype. Immunofluorescence analysis (Figure 1b) showed that after 21 days of culture time, AHCFs expressed cardiac fibroblast markers, such as vimentin and DDR2, and only 1.7% were positively stained for α-smooth muscle actin (α-SMA), suggesting that AHCFs retained their physiological phenotype. Then, ECM proteins were analyzed in pre- and post-decellularized samples. Nuclei staining using DAPI confirmed efficient cell removal in post-decellularized samples, as no nuclei could be detected compared to pre-decellularized samples (Figure 1c). Immunofluorescence analysis revealed production (Figure 1c, upper panel) and deposition of laminin and fibronectin (Figure 1c, lower panel) in pre- and post-decellularized samples. After decellularization, laminin and fibronectin were retained on the surface of PS dishes. Collagen I and collagen IV were also produced by AHCFs (Figure 1d, upper panel). After decellularization, collagen IV was retained on PS dishes, while only traces of collagen I could be detected (Figure 1d, lower panel). Overall, these results indicated the successful in vitro production of a cardiac tissue-like ECM which, after decellularization, was mainly composed of laminin, fibronectin and collagen IV, with no nucleic residues, referred to as cardiac BM.

### 3.2. AHCF Adhesion and Reprogramming on BM-coated PS Plates

After BM preparation and characterization, BM-coated PS dishes were analyzed for their ability to support AHCF adhesion compared to uncoated PS dishes. AHFCs were seeded on uncoated and BM-coated PS dishes (Figure 2a) and cell adhesion was analyzed at different time points (0.5, 1, 2 and 4 h) (Figure 2c). BM-coated PS dishes accelerated AHCF adhesion resulting in ~50% of adherent cells after 30 min, versus ~20% on PS substrates (*p* value = 0.042) (Figure 2c). However, both substrates showed complete AHCF adhesion after 4 h of culture time.

Then, BM-coated PS dishes were analyzed for their ability to support the direct reprogramming of miRcombo-transfected AHCFs, compared to uncoated PS plates. After transient transfection with miRcombo or negmiR (Figure 2b), cells were detached and seeded onto BM-coated plates, or left on the same uncoated PS dishes. AHCFs transfected with miRcombo expressed increased TNNT2 levels after 15 days of culture compared to negmiR-transfected cells cultured on uncoated (*p* value = 0.0004, Figure 2d) and BM-coated PS dishes. Interestingly, the expression of TNNT2 cardiomyocyte gene significantly increased for miRcombo-transfected AHCFs cultured on BM-coated PS dishes compared to uncoated plates (*p* value = 0.021), while TNNT2 expression was unaltered for negmiR-transfected cells (*p* value = 0.009). ACTC1 and CACNA1C expression was similar for miRcombo- and negmiR-transfected cells cultured on uncoated PS plates and negmiR-transfected cells cultured on BM-coated plates (Figure 2e,f). Meanwhile, BM-coated plates significantly increased ACTC1 and CACNA1C expression in miRcombo-transfected cells compared to control negmiR-transfected cells (*p* value = 0.046 and 0.002, respectively) and miRcombo-transfected cells cultured on PS dishes (*p* value = 0.0048 and 0.003, respectively) (Figure 2e,f).

Furthermore, flow cytometry analysis showed a significantly higher percentage of cardiac troponin T (cTnT)-positive cells (14.6%) for miRcombo-transfected AHCFs cultured for 15 days on BM-coated dishes compared to negmiR controls (Figure 2g) and to uncoated PS plates (~11% cTnT-positive cells as reported in our previous study [11]) (Figure 2h). Hence, BM coating was found to increase the direct reprogramming efficiency of miRcombo-transfected AHCFs into iCMs.

### 3.3. Physical Characterization of 3D Hydrogels

Fibrin-based hydrogels with/without BM (3D and 3D BM, respectively) were prepared and analyzed for their morphological and rheological properties. Scanning electron microscopy (SEM) was performed on freeze-dried hydrogel samples to analyze the effect of BM addition on hydrogel morphology. SEM images of the 3D hydrogel (Figure 3a left panel) showed the typical fibrillar structure of a fibrin hydrogel. The morphology of the 3D BM hydrogel section suggested the formation of a compatible blend with BM interaction with fibrin fibrils. The presence of BM in the 3D BM hydrogel was also confirmed by immunofluorescence staining of fibronectin, laminin and collagen IV (Figure 3b), while no fluorescence signal was detected in the 3D hydrogel (left panel), used as a control.

The rheological properties of 3D and 3D BM hydrogels were evaluated by frequency sweep analysis (Figure 3c). Both compositions showed a gel-like behavior with the storage modulus (G’) higher than the loss modulus (G’’) for the range of frequencies investigated. The elastic modulus at 1 Hz (Figure 3d) was obtained from the storage modulus as described in the materials and methods section, assuming a Poisson’s ratio of 0.5 [23,24]. Three-dimensional hydrogels showed an average elastic modulus of 64 ± 7 Pa, while an average value of 23 ± 1 Pa was obtained for 3D BM hydrogel samples. These results indicated than BM introduction in the fibrin-based hydrogels decreased the viscoelastic properties of the final materials (*p* value = 0.0005), leading to the formation of significantly softer structures.

### 3.4. miRcombo-Mediated Direct Reprogramming of AHCFs into iCMs in 3D BM Hydrogels

Fibrin-based hydrogels were evaluated as biomimetic culture systems to enhance miRcombo-mediated reprogramming of AHCFs into iCMs. miRNA-transfected cells were cultured either on the fibrin hydrogel’s surface (2D hydrogel) or embedded into the hydrogel (3D hydrogel) to decouple the effect of mechanical stiffness and composition from 3D culture microenvironment.

Flow cytometry analysis showed that the percentage of cTnT^+^ cells deriving from miRcombo-transfected cells cultured for 15 days on 2D hydrogels was higher (~14%) than on uncoated PS plates and for control negmiR-transfected cells on 2D hydrogels (Figure 4b,c). Interestingly, reprogramming efficiency evaluated as the percentage of cTnT^+^ cells was similar for cells cultured on 2D hydrogels and BM-coated PS plates. Then, the effect of the 3D culture condition (3D hydrogel) on the direct reprogramming efficiency was analyzed. DdPCR analysis showed that miRcombo-mediated direct reprogramming was more efficient when cells were cultured on 2D hydrogels than on PS plates, as evidenced by the increased expression of TNNT2 (*p* value = 0.048) (Figure 4d). However, miRcombo-transfected cells in 3D hydrogel showed increased expression of TNNT2 compared to their culture on 2D hydrogel (*p* value = 0.0057) and PS plates (*p* value = 0.002), whereas no significant increase in TNNT2 expression was observed in negmiR-transfected control cells (Figure 4d). Furthermore, miRcombo-transfected cells in 3D hydrogels showed increased expression of MLY7 and SCN5A (Figure 4e,f) compared with their cultures on 2D hydrogels (respectively *p* value = 0.003 and 0.047) and PS plates (respectively *p* value = 0.013 and 0.018). Such results demonstrate that cultures in 3D fibrin hydrogel strongly reinforced miRcombo-mediated reprogramming of AHCFs into iCMs, compared to 2D culture conditions.

To determine whether cardiac-specific proteins may further enhance miRcombo-mediated direct reprogramming in 3D culture conditions, transfected AHCFs were cultured in a BM-containing fibrin hydrogel to recreate a cardiac-specific 3D microenvironment. MiRcombo-transfected cells were encapsulated in fibrin/biomatrix hydrogels (3D BM hydrogel) 24 h post-transfection and cultured for 15 days. Then, the expression of different cardiomyocyte markers was analyzed to detect any gene expression change in cells cultured in 3D BM hydrogels compared to all previously analyzed miRcombo-transfected cell conditions. Overall, high expression of TNNT2, MYL7, SCN5A, ACTC1 and CACNA1C was found in 3D BM hydrogels. A significant increase in TNNT2 expression was found in 3D BM hydrogel compared to all conditions, thus demonstrating that increased TNNT2 expression is reached thanks to 3D culture in the presence of BM (Figure 4g). Considering the expression of SCN5A (Figure 4h) and MYL7 genes (Figure 4i), no significant differences in their expression were detected in 3D BM hydrogel when compared to 3D hydrogel. On the other hand, the expression of ACTC1 (Figure 4j) and CACNA1C genes (Figure 4k) increased in 3D BM hydrogel compared to PS, 2D hydrogel and 3D hydrogel, but no significant differences were evidenced compared to cultures on BM-coated PS dishes.

Immunofluorescence analysis showed cTnT expression in miRcombo-transfected cells cultured in 3D and 3D BM hydrogels for 15 days, with a percentage of cTnT-positive cells of around 40% and 50%, respectively (Figure 5a,b).

Moreover, we assessed the Ca^2+^ handling of miRcombo-transfected cells cultured in 3D and 3D BM hydrogels for 15 days. Culturing in 3D BM hydrogel caused an increase in the percentage of cells showing Ca^2+^ transient compared to 3D hydrogels (32% vs. 20%, respectively) (Figure 5c), and caused more rapid Ca^2+^ release upon caffeine stimulation, suggesting an enhancement in functionality (Figure 5d,e).

Such results demonstrate that the route towards the direct reprogramming of miRcombo-treated AHCFs into iCMs may be significantly enhanced by culturing the cells in a biomimetic 3D BM hydrogel.

## 4. Discussion

Since the first report published in 2010 [4], direct reprogramming of fibroblasts into iCMs has been extensively investigated for cardiac regenerative medicine. Previous studies have mainly focused on the optimization of the reprogramming agents (selected among cardiac TFs, small molecules and miRNAs) to obtain iCMs in vitro from mouse (embryonic, tail-tip or neonatal fibroblasts) fibroblasts [25,26]. However, the translation of reprogramming approaches from mouse to human adult cells has generally failed to efficiently generate iCMs due to interspecies differences [27]. Additional factors have been generally required to address specific barriers to the direct reprogramming of human adult fibroblasts into iCMs [27,28,29]. In particular, microenvironmental biochemical and biophysical cues represent poorly explored stimuli with the potentiality to enhance direct reprogramming efficiency and iCM maturation level [12]. Indeed, sarcomere proteins are generally not well organized in human iCMs, while the obtainment of in vitro spontaneously beating human iCMs has been rarely reported [12]. Interestingly, in vivo studies in mouse models have shown the formation of iCMs with higher efficiency and maturation level than in vitro experiments, with similar transcriptome, structure and functions of endogenous cardiomyocytes [30]. The improvement of direct reprogramming outcomes in vivo has been attributed to the presence of complex combinations of tissue-specific stimuli in the cardiac tissue including a 3D structure of cardiac ECM providing biophysical and biochemical cues to cells, paracrine factors secreted by close-by cells and cell–cell contacts in a 3D environment. In our previous study, miRcombo-single transient transfection was successful in triggering human AHCF reprogramming into iCMs on uncoated PS culture plates [11]. However, direct reprogramming efficiency was limited, sarcomere structures were not observed within iCMs, and no beating cells were present up to 30 days post-transfection. Hence, this study suggested the need for additional agents or specific culture conditions aimed at improving direct reprogramming outcomes.

Based on the encouraging results of previous in vivo trials, in this study, we investigated the ability of a 3D cardiac tissue-like culture environment in enhancing miRcombo-mediated direct reprogramming of AHCFs into iCMs. To this end, the biochemical stimuli of cardiac tissue-like ECM and the biophysical stimuli exerted by a 3D fibrin hydrogel were first explored and then synergistically combined.

Previously, laminin and RGD have been integrated into PEG hydrogels to enhance the direct cardiac reprogramming of mouse fibroblasts into iCMs [20]. Alternatively, Matrigel has been used as a coating in polyacrylamide hydrogels [14] or added into fibrin hydrogels [15] for studying the direct reprogramming efficiency of mouse fibroblasts into iCMs. However, Matrigel is a reconstitute basement membrane rich in laminin and collagen IV, derived from Engelbreth–Holm–Swarm mouse tumors [31]. Hence, Matrigel does not fully resemble the physiological microenvironment of human cardiac tissue due to its derivation from mouse tumors. Furthermore, its preparation is also not compliant with the 3Rs (Reduction, Replacement, Refinement) principle. In regenerative medicine, alternative ECM materials to Matrigel are highly in demand. A cardiac tissue-like ECM was herein produced by the 21-day culture of AHCFs followed by decellularization (BM; Figure 1a) [21]. Cardiac fibroblasts have a structural and functional role within the heart [32]. Among their different functions, cardiac fibroblasts are involved in ECM synthesis, deposition and remodeling, thus maintaining tissue homeostasis in physiological and pathological conditions [32]. Normal cardiac fibroblasts are defined by the expression of several markers, such as vimentin and DDR2, among other cytoskeletal and surface markers [33]. Conversely, α-SMA is a cytoskeletal protein, under the tumor growth factor-ꞵ (TGF-ꞵ) pathway’s control, expressed only by myofibroblasts that populate fibrotic hearts, such as scar tissue after MI [33]. In this work, AHCFs retained their phenotypic characteristics after 21 days of culture in vitro during BM deposition. Cardiac fibroblast activation into myofibroblasts was reported to be dependent on cell density in vitro, with increased α-SMA expression by culturing fibroblasts at low density [34]. These results suggest that AHCFs need cell–cell interactions to retain their normal phenotype [10]. As in this work a high AHCF density was used, reaching total cell confluency 48 h post cell seeding, fibroblast activation was not observed, despite the long culture time onto stiff PS plates, with only 1.67% of α-SMA-positive cells after 21 days of culture time (Figure 1b).

Protein composition of BM before and after decellularization was then investigated. BM was positively stained for fibronectin, laminin and collagen IV in post-decellularized samples, and no nuclei signals were detected in post-decellularized samples (Figure 1c,d). However, fluorescence intensity associated with collagen I’s presence was less intense in post-decellularized compared to pre-decellularized BM (Figure 1). Hence, in vitro-produced BM was mainly composed of fibronectin, laminin and collagen IV (Figure 1c,d), which, together with other proteins and molecules, constitute the cardiac basement membrane [17,35]. During heart development, these proteins recreate a supportive environment for cell proliferation and polarization and enclose cardiomyocytes in adult tissue, supporting their functions [17,35]. Previously, Zhen et al. reported the preparation of cardiac ECM in vitro from neonatal cardiac rat fibroblasts by a similar protocol, showing the presence of collagen fibers (of unspecified type) and complete removal of nuclei after decellularization [36]. Castaldo et al. reported the presence of fibronectin, laminin and collagen IV after the decellularization and upregulation of collagen I in pathological BM produced by activated fibroblasts [21]. Therefore, the limited presence of collagen I in BM could be due to its physiological nature, being produced from non-activated AHCFs. Indeed, collagen I is more abundant in infarcted tissue, increasing its stiffness [37]. In vitro-produced human cardiac BM was herein proposed as a tissue- and specie-specific bioactive material for use in hydrogels/scaffolds for cardiac regenerative medicine. Indeed, while animal-derived cardiac ECM can be produced by means of heart tissue decellularization, human cardiac ECM cannot be obtained using the same approach due to the scarce availability of human heart samples.

After BM characterization, we investigated BM’s involvement in AHCF adhesion and direct reprogramming. Firstly, AHCF adhesion on BM-coated PS plates was analyzed compared to uncoated PS dishes (Figure 2c). Interestingly, BM promoted cell adhesion in the first 30 min post cell seeding, reaching almost 50% of adherent cells. These results are in agreement with previous studies, showing increased cell adhesion on ECM protein-coated substrates compared to those that were uncoated [20]. Different studies have reported that fibroblasts undergoing direct reprogramming into iCMs start to express cardiac ECM proteins, such as fibronectin, collagen and laminin, in the first hours/days after reprogramming induction [7,19]. In this work, we investigated for the first time direct reprogramming of AHCFs into iCMs on an in vitro-produced human BM mimicking human cardiac ECM. After 15 days of culture on BM-coated PS dishes, miRcombo-transfected cells showed increased expression of TNNT2, ACTC1 and CACNA1C genes compared to cells cultured on control PS dishes (Figure 2d–f). Moreover, no differences in ACTC1 and CACNA1C gene expression were observed between miRcombo- and negmiR-tranfected cells cultured on PS dishes, suggesting that miRcombo-transfection alone is not able to induce the expression of certain CM genes, and cell interaction with BM is required. Moreover, the higher percentage of cTnT^+^ cells on BM-coated with respect to uncoated PS plates (~14% vs. ~11% [11]) suggested that cell binding to BM may enhance iCM generation (Figure 2g,h). Smith et al. showed that direct reprogramming of mouse fibroblasts on laminin-coated PS dishes increased the number of beating cells compared to uncoated PS after 18 days of culture [20]. Sa et al. reported ECM protein involvement in the maturation of cardiomyocytes, differentiated from human embryonic stem cells [38]. In detail, stem cell culture on PS dishes coated with a fibronectin/laminin mixture (70/30 wt./wt.) enhanced their differentiation into cardiomyocytes compared to a fibronectin, laminin and gelatin coating alone [38]. Blocking cell receptors for fibronectin and laminin reduced the differentiation potential into cardiomyocytes, suggesting the key role of such proteins in cardiomyocyte formation and maturation in vitro.

The biophysical properties of soft hydrogel matrices could also influence cell reprogramming efficiency [13]. Fibrin hydrogels are soft materials with viscoelastic properties depending on the concentration of fibrinogen and thrombin. In this work, fibrin hydrogels were produced from a fibrinogen solution with a 5 mg/mL final concentration containing 50 units/mL of thrombin (Figure 3c). Such hydrogels were soft, with an average storage modulus of 20 Pa (Figure 3d), in agreement with previous reports [39,40]. The addition of BM to fibrin hydrogels further decreased their viscoelastic properties (Figure 3c). SEM analysis of fractured sections of freeze-dried 3D hydrogels demonstrated that the fibrillar structure of fibrin was retained in the presence of BM, while phase separation was not observed, indicating the formation of a compatible blend (Figure 3a). A few studies investigated the effect of the incorporation of decellularized ECM on the rheological properties of fibrin hydrogels, obtaining different results. Williams et al. studied the combination of fibrin with decellularized ECM from rat ventricular tissue. No differences were detected between the mechanical properties of hybrid hydrogels composed of 3.3 mg/mL of fibrinogen with 340 µg/mL of ECM (corresponding to a final ECM content of 9.3% *w*/*w*) and matrices based on decellularized ECM only [41]. On the contrary, Jorgensen et al. investigated the use of hydrogels composed of decellularized skin ECM and fibrin as bioinks for 3D printing. In their study, an enhancement of the storage and loss moduli values was obtained for hybrid gels composed of 30 mg/mL of fibrinogen and 3 mg/mL of ECM (corresponding to a final ECM content of 9% *w*/*w*) compared to fibrin samples [42]. These discrepancies are probably associated with the different compositions of the decellularized tissue investigated, specifically the type and content of collagen present in the samples after the decellularization process. Furthermore, differences in hydrogel concentration and fibrin/decellularized ECM ratio could also be responsible for different hydrogel stiffness values. Recently, Kurotsu et al. reported that direct reprogramming of mouse fibroblasts into iCMs was enhanced when cells were cultured on Matrigel-coated polyacrylamide hydrogels with 8 kPa stiffness compared to Matrigel-coated hydrogels with 126 kPa stiffness and Matrigel-coated PS dishes (stiffness of ~1 GPa) [14]. Hence, higher direct reprogramming efficiency was achieved for cells cultured on substrates with stiffness close to that of physiological cardiac tissue (~10 kPa), and YAP/TAZ inhibition was identified as the main responsible molecular mechanism. A limitation of this report relies on the impossibility to decouple biophysical signaling of polyacrylamide hydrogels from biochemical signaling of Matrigel, due to the non-adhesive nature of polyacrylamide hydrogels. As Matrigel is rich in laminin and collagen IV, providing biochemical stimuli to cells, it probably affected direct reprogramming results [31]. In contrast with the above report, in our study, miRcombo-transfected cells on 2D fibrin hydrogels (with around 65 Pa stiffness) showed a percentage of 15% cTnT-positive after 15 days culture, significantly higher than that measured for miRcombo-transfected cells cultured on stiffer PS dishes (~11% [11]) (Figure 4b,c). In the case of fibrin hydrogel, a soft matrix was appropriate for improving direct reprogramming efficiency of AHCFs into iCMs. Combined with previous findings by Kurotsu et al. [14], it appears evident that efficient iCM generation from AHCFs requires cell culturing on softer matrices than PS dishes.

Next, we investigated cell reprogramming in 3D fibrin hydrogel culture systems. MiRcombo-mediated reprogramming of murine fibroblasts into iCMs has been previously studied culturing cells into 3D fibrin/Matrigel hydrogels [15]. Cells exhibited an increased reprogramming efficiency when cultured in 3D hydrogels compared to PS dishes. However, in this previous report, reprogramming efficiency was not evaluated for miRcombo-transfected cells cultured on the surface of hydrogels, and uncoated PS dishes were used as the only control culture condition. Moreover, the employed 3D hydrogel was a mixture of fibrin and Matrigel, for which biochemical stimuli can also affect direct reprogramming efficiency. In our study, we demonstrated that miRcombo-mediated direct reprogramming of AHCFs is increased in 3D hydrogels compared to PS and 2D hydrogel conditions (Figure 4d–f). MiRcombo-transfected cells exhibited increased expression of TNNT2, MYL7 and SCN5A compared to negmiR-transfected cells cultured in 3D hydrogels, and to all cells cultured in 2D conditions, suggesting a key role of 3D culture substrates in direct cell reprogramming (Figure 4d–f).

Finally, we investigated the direct reprogramming of AHCFs cultured in a biomimetic 3D microenvironment based on a 3D fibrin-based hydrogel containing BM, combining biochemical and biophysical cues. With respect to fibrin hydrogels, mechanical stiffness of 3D BM hydrogels decreased to around 23 Pa (Figure 3d). However, direct reprogramming outcomes for cells cultured in 3D BM hydrogels were not significantly affected by this change in mechanical properties, but rather they were the result of combined effects by 3D fibrin hydrogel and BM addition. Indeed, by introducing BM into 3D hydrogels, miRcombo-transfected cells exhibited increased reprogramming efficiency compared to other tested culture conditions. Interestingly, expressions of TNNT2 gene and cTnT protein were increased for cells cultured in 3D BM hydrogels compared to 3D hydrogels without BM (Figure 4g,l). Conversely, no significant differences were observed in gene expression levels of MYL7 and SCN5A genes (Figure 4h,i) for cells in 3D culture conditions with/without BM, thus demonstrating that 3D culture was responsible for the regulation of the expression of such genes. In contrast, ACTC1 and CACNA1C gene expression was regulated by BM’s presence rather than the 3D fibrin hydrogel (Figure 4j,k). Increased cell function of miRcombo-cell reprogrammed in 3D BM hydrogel was also observed when performing calcium transient analysis (Figure 5c–e). Overall, these results demonstrate that 3D culture in a soft fibrin hydrogel containing human BM significantly increased miRcombo-mediated reprogramming of AHCF into iCMs. In the future, the molecular mechanisms responsible for changes in the direct reprogramming efficiency of AHCs in iCMs based on the culture microenvironment will be elucidated, analyzing the expression of matrix metalloproteinases [15] and YAP/TAZ signaling [14], as previously suggested.

As a conclusion, this report shows that tailor-designed biomimetic substrates may help to overcome the molecular barriers to direct reprogramming of AHCFs into iCMs. This research paves the way to further interdisciplinary studies aimed at knowledge advancements for future translation of this technology into the clinics.

## 5. Conclusions

MiRcombo-mediated direct reprogramming of fibroblasts into iCMs has emerged as a potentially promising cardiac regenerative approach after myocardial infarction. However, efficient translation of this technology into clinics still requires extensive investigations aimed at improving its efficiency. In this work, we elucidated the synergistic effect of cardiac tissue-like biochemical and biophysical stimuli in enhancing the direct reprogramming efficiency of AHCFs into iCMs. Biochemical stimuli were provided by BM, an in vitro-produced human cardiac tissue-like ECM, while biophysical stimuli were imparted by 3D culture in a soft fibrin-based hydrogel. The culturing of miRcombo-transfected AHCFs in 3D BM hydrogels showed a synergistic effect, significantly increasing the expression of different cardiac genes in miRcombo-transfected cells. Future studies will elucidate the molecular mechanisms activated by the cardiac tissue-like culture microenvironment. Furthermore, this research work will stimulate further interdisciplinary research aimed at enhancing direct cardiac reprogramming outputs for its closer translation into clinics.

## Figures and Tables

**Figure 1 cells-11-00800-f001:**
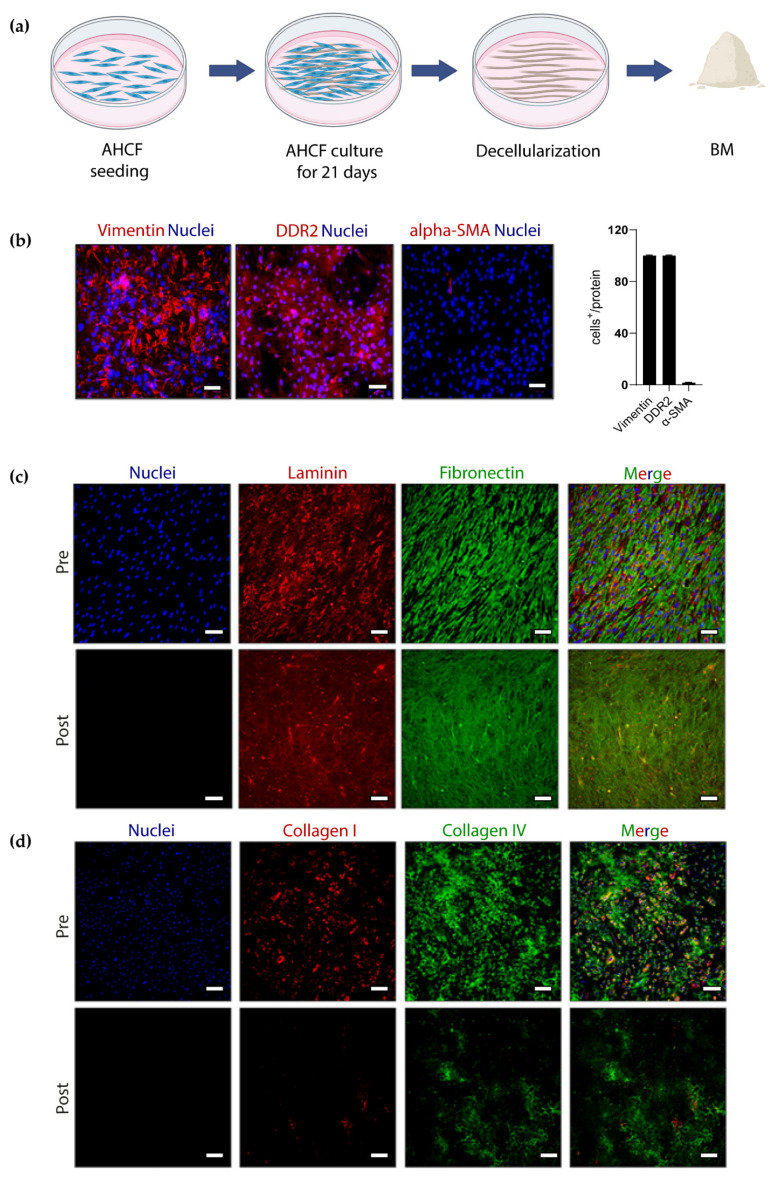
BM production and characterization in vitro: (**a**) schematic representation of BM production in vitro by AHCFs. Image created with Biorender.com under license. (**b**) Immunofluorescence analysis and quantification of AHCFs expressing vimentin, DDR2 and α-SMA after 21 days of culture (*n* = 3). Nuclei were counterstained using DAPI. Scale bar: 100 µm. (**c**,**d**) Immunofluorescence analysis of ECM proteins laminin, fibronectin, collagen I and IV before (upper panel, *n* = 3) and after (lower panel, *n* = 3) decellularization at 21 days of culture. Nuclei were counterstained using DAPI. Scale bar, 100 µm.

**Figure 2 cells-11-00800-f002:**
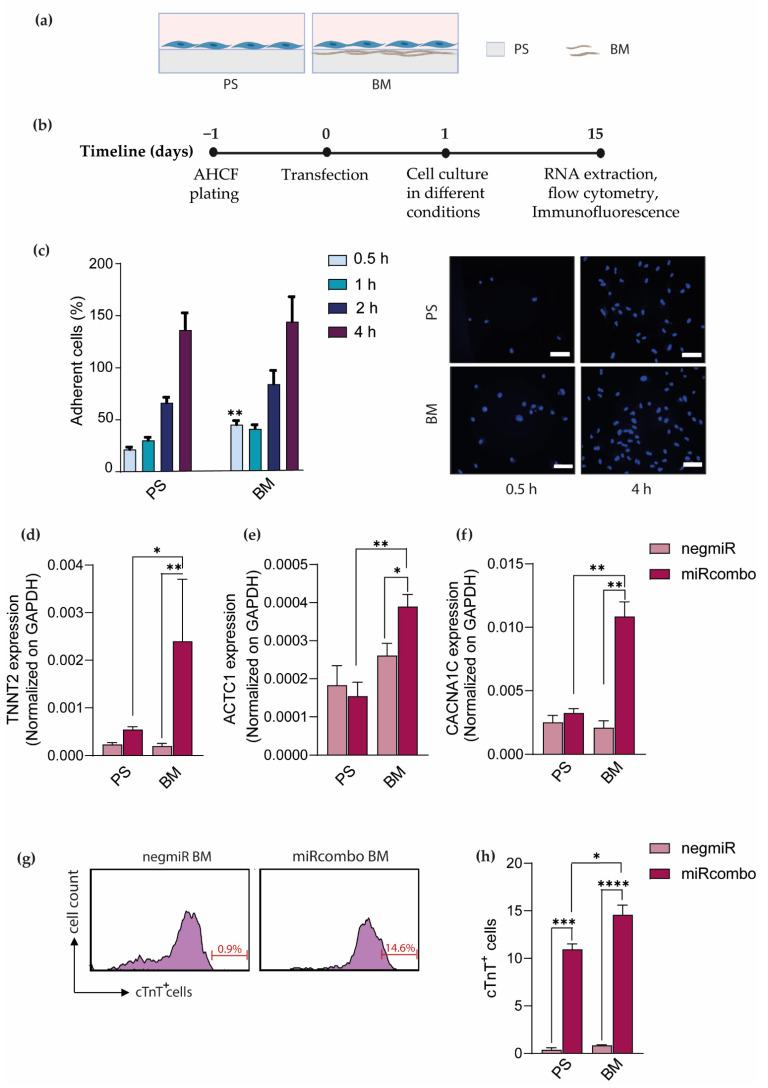
BM-coated PS plates enhanced AHCF adhesion and miRcombo-mediated direct reprogramming into iCMs. Schematic representations of: (**a**) cell culture conditions on uncoated polystyrene (PS) and BM-coated PS dishes; (**b**) cell transfection (with miRcombo or negmiR) and culture on different conditions for 15 days post-transfection. Image created with Biorender.com under license. (**c**) Cell adhesion on uncoated and BM-coated PS dishes at different time points (0.5, 1, 2 and 4 h): cell adhesion percentage (left panel) and representative images (right panel) of adherent DAPI-stained cells. Experiments were conducted in triplicate, five fields for each sample were taken using a fluorescence microscope. Nuclei were counterstained using DAPI. Scale bar, 100 µm. (**d**–**f**) Expression of TNNT2, ACTC1 and CACNA1C cardiomyocyte genes analyzed by ddPCR in AHCFs transfected with negmiR or miRcombo and cultured on uncoated or BM-coated PS plates for 15 days. Results, expressed as DNA copies/µL, are the average of three independent experiments, each performed in triplicate. (**g**) Representative flow plots of cTnT^+^ cells in AHCFs transfected with negmiR or miRcombo and cultured on BM-coated PS plates (*n* = 3) for 15 days. (**h**) Graph showing flow cytometry percentage of cTnT^+^ cells in AHCFs transfected with negmiR or miRcombo and cultured on PS or BM-coated PS plates (*n* = 3) for 15 days. All data are expressed as the mean ± SEM. Statistical differences between the groups were determined with two-sided *t*-tests. * *p* < 0.05, ** *p* < 0.01 and *** *p* < 0.001, **** *p* < 0.0001.

**Figure 3 cells-11-00800-f003:**
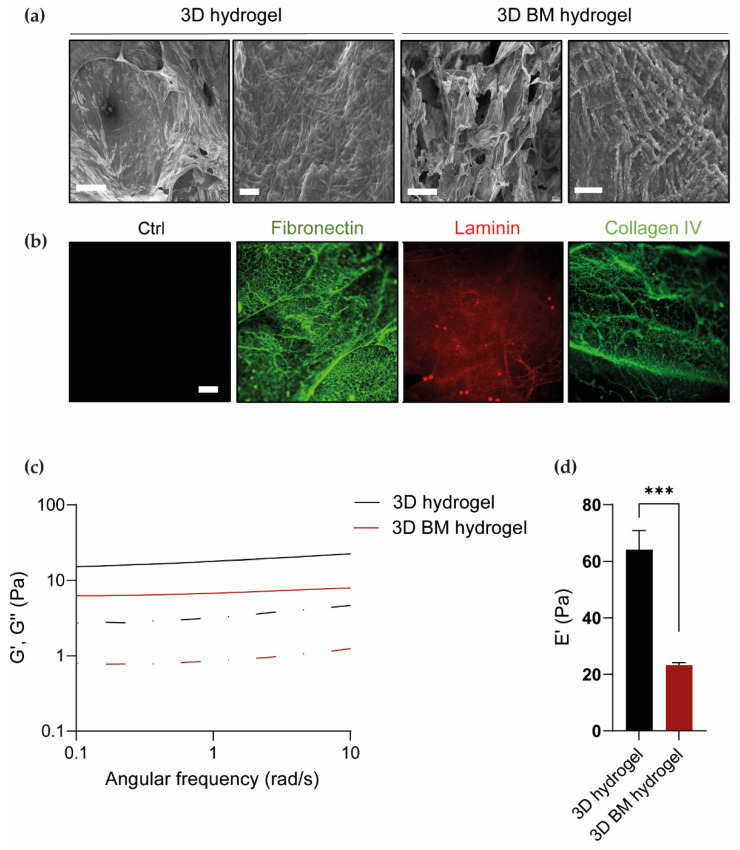
Three-dimensional and 3D BM hydrogel characterization. (**a**) SEM images of 3D hydrogel and 3D BM hydrogel sections at different magnifications (left panel scale bar = 100 µm; right panel scale bar = 1 µm). (**b**) Immunofluorescence of the ECM proteins fibronectin, laminin and collagen IV in 3D BM hydrogels (*n* = 3). The 3D hydrogel without BM was used as a control. Scale bar = 100 µm. (**c**) Storage modulus (G’, continuous line) and loss modulus (G”, dotted line) of 3D hydrogel and 3D BM hydrogel as a function of angular frequency (*n* = 3). (**d**) Elastic modulus (Pa) of 3D hydrogel and 3D BM hydrogel (*n* = 3). All data are expressed as the mean ± SEM. Statistical differences between the groups were determined with two-sided t-tests. All *p* values (*** *p* < 0.001) to determine statistical significance are presented in the charts.

**Figure 4 cells-11-00800-f004:**
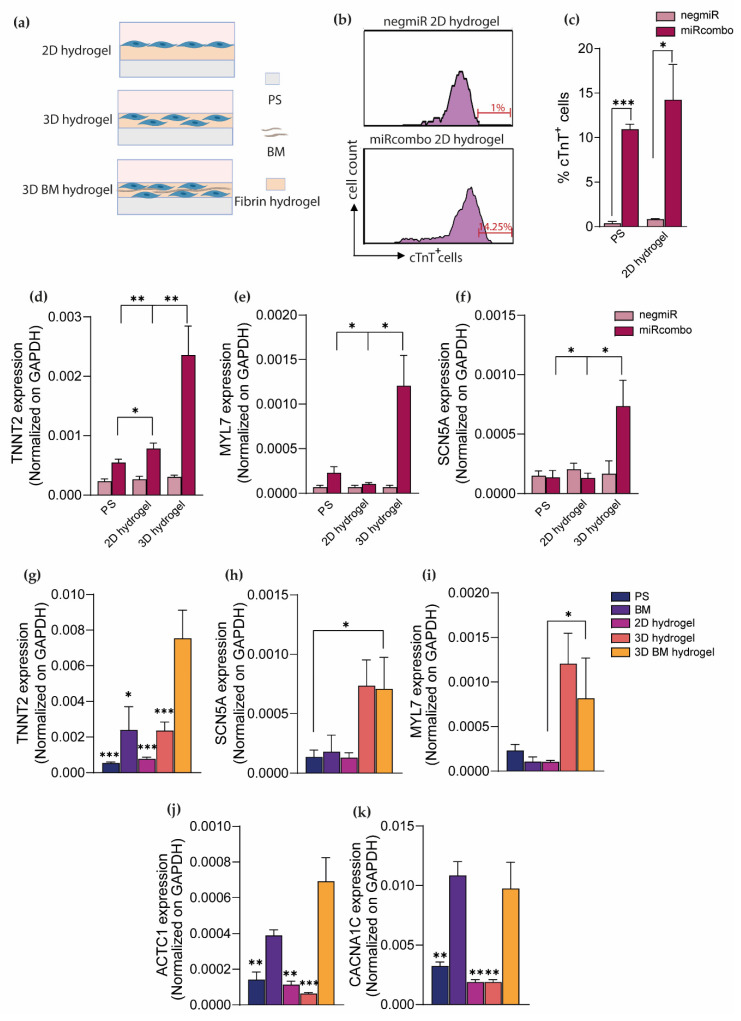
Three-dimensional BM hydrogel enhances miRcombo-mediated cell reprogramming as assessed by the analysis of cardiac marker expression by ddPCR analysis and flow cytometry. (**a**) Representative images of different culture conditions: transfected cells are cultured on the top of previously formed hydrogels (2D hydrogel), encapsulated into 3D hydrogel (3D hydrogels) and into 3D hydrogel with BM (3D BM hydrogel)**.** Image created with Biorender.com under license. (**b**) Representative flow plots of cTnT^+^ cells from AHCFs transfected with negmiR or miRcombo on 2D hydrogel (*n* = 3) and cultured for 15 days. (**c**) Graph showing flow cytometry percentage of cTnT^+^ cells from AHCFs transfected with negmiR or miRcombo on PS and 2D hydrogel (*n* = 3) and cultured for 15 days. (**d**–**f**) ddPCR analysis of TNNT2, MYL7 and SCN5A cardiomyocyte gene expression in AHCFs transfected with negmiR or miRcombo and cultured on PS plates, 2D and 3D hydrogels for 15 days. All data are expressed as the mean ± SEM. Statistical differences were determined with two-sided t-tests. (**g**–**k**) ddPCR analysis of TNNT2, MYL7, SCN5A, ACTC1 and CACNA1C cardiomyocyte genes for miRcombo-transfected cells cultured in PS, BM, 2D hydrogels, 3D hydrogels and 3D BM hydrogels (*n* = 3) for 15 days. All data are expressed as the mean ± SEM. Statistical differences are reported between 3D BM hydrogels compared to other culture conditions and were determined with two-sided t-tests. * *p* < 0.05, ** *p* < 0.01 and *** *p* < 0.001.

**Figure 5 cells-11-00800-f005:**
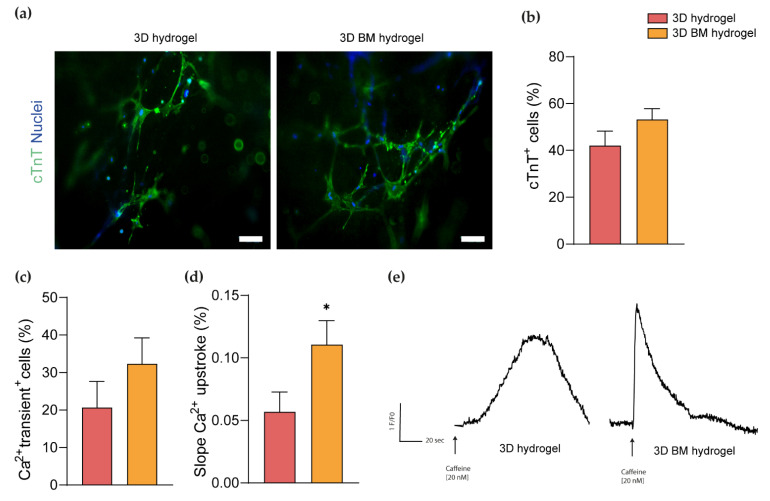
Three-dimensional BM hydrogel enhances miRcombo-mediated cell reprogramming as assessed by immunofluorescence and calcium transient analysis. Immunofluorescence analysis: (**a**) cTnT expression and (**b**) percentage of cTnT-positive cells in miRcombo-transfected AHCFs cultured in 3D and 3D BM hydrogels for 15 days. Blue: nuclei counterstained using DAPI. Green: cTnT. Scale bar: 50 µm. (**c**) Percentage of cells showing Ca^2+^ transients, (**d**) slope of Ca^+2^ upstroke after caffeine induction (* *p* < 0.05) and (**e**) trend of Ca^+2^ oscillation vs. time after caffeine induction, for miRcombo-transfected cells in 3D and 3D BM hydrogels after 15 days culture time.

## Data Availability

The datasets generated for this study are available on request to the corresponding author.

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
