# Peer review of "Cardiac Tissue-like 3D Microenvironment Enhances Route towards Human Fibroblast Direct Reprogramming into Induced Cardiomyocytes by microRNAs"

_cells, 2022, doi:10.3390/cells11050800_

Round 1

Reviewer 1 Report

In this paper, the author explored the microenvironment enhances human fibroblast direct reprogramming. Although the concept is novel, there is no sufficient data to the author made a successful reprogramming.

Major issues:

  1. In figure 4b, I cannot find the cTnT+ cells from the FACS result.
  2. In figure 4d-4f, I cannot really believe there is reprogramming based on ddPCR results. The author only makes the gene expression, I don’t know what is that means. If the gene expression level is only 0.0005, I don’t think this gene really expressed. Also, the author also needs to show the fibroblast gene expression level decreased.
  3. In figure 4I, the author showed the immunofluorescence images of cTnT in 3D and 3D BM hydrogels group, but I cannot believe there is cTnT positive cell only based on a low magnification image, the author needs to show a higher magnification image to show the sarcomere structures of the reprogramming. Actually, the whole paper hasn’t shown any iCM image to prove there is successful reprogramming.
  4. In the paper, the author mentioned that the MYL7 expression level is higher in 3D hydrogels than in 2D conditions. As I know that Myl7 is an atrial marker, does that mean the 3D hydrogels will generation more atrial cardiomyocytes? The author needs to check ventricular, and pacemaker cardiomyocytes gene expression level by ddPCR or Immunofluorescence staining.
  5. If the reprogramming is true and the 3D-BM hydrogels really enhances human fibroblast direct reprogramming, the author needs to use a better way to present the result.

Minor issues:

  1. Please list the primer sequence for ddPCR.
  2. Please provide the catalog number of the antibodies used in the paper.
  3. In the paper, the author mentioned that the stiffness of 2D fibrin hydrogels with around 65 Pa stiffness, so is that possible the measure the stiffness of BM-coated plate? One more question, does the author notice the difference in the stiffness of the substrate between mouse and human reprogramming? Mouse always need around 8-10 kilopascals, but human cells only need less than 100 pascals.

Author Response

Cardiac tissue-like 3D microenvironment enhances human fibroblast direct reprogramming into induced cardiomyocytes by microRNAs

Camilla Paoletti1,2,*, Elena Marcello1,2 , Maria Luna Melis1,2 , Carla Divieto3 , Daria Nurzynska4 and Valeria Chiono1,2

Response to Reviewers

We would like to express our gratitude to the Reviewers for their careful assessment, which allowed us to improve the quality of this paper. The Reviewers’ comments were carefully addressed: corresponding changes in the manuscript were implemented and highlighted in the revised version of the manuscript.

We have modified the manuscript’s title in: Cardiac tissue-like 3D microenvironment enhances route towards human fibroblast direct reprogramming into induced cardiomyocytes by microRNAs

Detailed answers to Reviewers’ comments are reported below.

Reviewer 1

Major issues:

  1. In figure 4b, I cannot find the cTnT+ cells from the FACS result.

Figure 4b reports cTnT positive cells for 2D hydrogels after negmiR (1%) miRcombo treatment (14%). Graph in 4c reports the percentage of cTnT positive cells on 2D hydrogels compared polystyrene tissue culture plates (Paoletti et al, Frontiers in Bioengineering and Biotechnologies 2020 https://doi.org/10.3389/fbioe.2020.00529)

  1. In figure 4d-4f, I cannot really believe there is reprogramming based on ddPCR results. The author only makes the gene expression, I don’t know what is that means. If the gene expression level is only 0.0005, I don’t think this gene really expressed. Also, the author also needs to show the fibroblast gene expression level decreased.

The expression levels indicated in the graph are obtained after normalization on GAPDH which is highly expressed. If we consider the ranges of absolute copy number/µL of the cardiomyocyte genes, they are around 50 copies/µL. This concentration is clearly a positive results of gene expression, considering that the ddPCR has a much lower limit of detection, less than 0,5 copies/µL.

Moreover, our results on sodium and calcium channel gene expression are in agreement with previous findings reported by Jayawardena et al. (DOI: 10.1161/CIRCRESAHA.112.269035), the first group demonstrating  miRcombo-mediated direct reprogramming in vitro and in vivo. Jayawardena et al showed (in their figure 2D) that miRcombo transfection enhances the expression of SCN5A and CACNA1C and that sodium channel expression is much lower (although significantly higher compared to negmiR control) than calcium channel gene expression.

Regarding fibroblast markers, we have already reported that miRcombo transient transfection significantly reduces fibroblast marker expression: we analysed Vimentin, Discoidin domain receptor-2 (DDR2) and Fibroblast specific protein-1 (FSP-1), 15 days after transfection compared to negmiR controls (Paoletti et al, Frontiers in Bioengineering and Biotechnologies 2020 https://doi.org/10.3389/fbioe.2020.00529). In the present study, we analysed fibroblast marker expression in all different culture conditions. We observed downregulation of fibroblast marker expression in miRcombo treated cells compared to negmiR controls, however we did not observe any significant differences between all miRcombo-tested conditions (PS, BM, 2D, 3D hydrogel and 3D BM hydrogel). This result suggested that the employed culture conditions did not affect fibroblast marker expression, which on the other hand seemed mainly affected by miRcombo. However the increase in cardiac marker expression for miRcombo-treated cells in 3D BM hydrogel suggested that this culture condition promoted the route toward direct reprogramming of AHCFs into iCMs.

  1. In figure 4I, the author showed the immunofluorescence images of cTnT in 3D and 3D BM hydrogels group, but I cannot believe there is cTnT positive cell only based on a low magnification image, the author needs to show a higher magnification image to show the sarcomere structures of the reprogramming. Actually, the whole paper hasn’t shown any iCM image to prove there is successful reprogramming.

In Figure 5 a-b), immunofluorescence analysis results are reported and the analysis of the percentage of cTnT positive cells was added as assessed by Immunofluorescence, counting the percentage of cTnT positive cells respect to total nuclei.

This additional analysis shown in Fig 5b) confirmed the expression of cTnT which was slightly higher for miRcombo-treated cells cultured in the 3D BM hydrogel.

The sarcomeric structure could not be clearly detected after only 15 days culture time, in agreement with previous studies from Li et al (doi.org/10.1038/srep38815), using mouse fibroblasts, after 15 day of culture. Sarcomeric organization using human fibroblasts is hard to obtain in vitro and requires prolonged culture, as assessed by Wang et al (https://doi.org/10.1016/j.biomaterials.2021.121028), where cTnT organization using human fibroblasts was evaluated after 4 weeks of culture.

The manuscript is aimed at demonstrating that biomimetic 3D culture conditions can enhance the route toward direct cell reprogramming of adult human cells into iCMs. The results of this work suggest that biomimetic 3D culture systems could be exploited as tools for enhancing in vitro direct reprogramming. While we did not obtain mature cardiomyocytes in 15 days of culture from AHCFs, our results are extremely innovative and promising for any further advancement in the field.

  1. In the paper, the author mentioned that the MYL7 expression level is higher in 3D hydrogels than in 2D conditions. As I know that Myl7 is an atrial marker, does that mean the 3D hydrogels will generation more atrial cardiomyocytes? The author needs to check ventricular, and pacemaker cardiomyocytes gene expression level by ddPCR or Immunofluorescence staining.

We thank the Reviewer for this comment. In this study, we have analysed the expression of markers that are typically investigated in fibroblast reprogramming into induced cardiomyocyte, spanning from ion channel gene, sarcomeric and functional gene expression, as reported by Jayawardena et al (DOI: 10.1161/CIRCRESAHA.112.269035), Li et al (doi.org/10.1038/srep38815), Wang et al (https://doi.org/10.1016/j.biomaterials.2021.121028). The aim of this study was to assess if reprogramming efficiency is increased using this innovative culture method based on 3D hydrogel and Biomatrix. In further study, it will be interesting to evaluate if cell reprogramming is directed to a specific iCM population.

  1. If the reprogramming is true and the 3D-BM hydrogels really enhances human fibroblast direct reprogramming, the author needs to use a better way to present the result.

 We thank the Reviewer for this comment. We added calcium transient studies in the paragraph 3.4, showing that culture in 3D BM hydrogels increased the percentage of miRcombo-treated cells showing Ca+2 transients, with increased slope of Ca+2 upstroke compared to miRcombo-treated cells cultured in 3D hydrogels (without BM).

Minor issues:

  1. Please list the primer sequence for ddPCR.

We thank the reviewer for this comment. Our primers are all supplied by Bio-Rad. We have added ID assay for each primer, which directly links to Bio-Rad primer details with all technical specification, including primer sequences.

  1. Please provide the catalogue number of the antibodies used in the paper.

We thank the reviewer for this comment. We added the catalogue number for antibodies in the materials and methods paragraph

  1. In the paper, the author mentioned that the stiffness of 2D fibrin hydrogels with around 65 Pa stiffness, so is that possible the measure the stiffness of BM-coated plate? One more question, does the author notice the difference in the stiffness of the substrate between mouse and human reprogramming? Mouse always need around 8-10 kilopascals, but human cells only need less than 100 pascals.

We thank the Reviewer for this comment. We did not measur the stiffness of BM-coated plate. BM-coated plates stiffness is very close to tissue culture plates without coating (~ 1 GPa), as coating solution concentration was just 100 µg/mL and we used 300, 200 and 100 µl of BM solution for each plate (12-, 24- and 96- multiwell plates, respectively).

We did not perform experiments on mouse fibroblasts. However, enhanced miRcombo-mediated reprogramming of mouse fibroblasts into iCMs using 3D hydrogels has been reported by Li et al (doi.org/10.1038/srep38815). This study showed that mouse fibroblast reprogramming was enhanced using a 3D fibrin-based hydrogel, formed by 1 mg/mL of fibrinogen, together with Matrigel. We carried our rheological analyses on fibrin hydrogels with different fibrinogen concentrations and/or thrombin content (data not published) which showed that fibrin hydrogels prepared from 1 mg/mL fibrinogen (as those used by Li et al.) have a low stiffness of ~25 Pa similar to that of our hydrogel prepared from 5 mg/mL. Furthermore, it is important to notice that up to now the scientific literature has always considered the effect of INITIAL mechanical properties (with reference to stiffness only as a mechanical parameter) on cardiac direct reprogramming. As different hydrogels have distinct viscoelastic properties which also change with time (due to cell remodelling), while state of the art data are limited to only a few papers, any detailed comparison between literature data on mouse cell reprogramming and our results on human cell reprogramming is currently not possible and more systematic studies would be needed.

Reviewer 2 Report

This article describes the improvement of cardiomyocyte regeneration efficiency by microenvironmental strategies. The authors and their research team provide a detailed background and discussion of the experiments, and the steps and materials used to perform the experiments are clearly described. However, if some of the concerns could be further explained, it is believed that the article would be further enhanced.
Major concern:

  1. the authors only used cTnT staining to claim that it is induced cardiomyocyte, which we believe may not be strong enough. Since the authors' team has the ability to culture and confirm the physiological nature of cardiomyocytes (Paoletti et al., 2020), comparing the physiological and biochemical properties of cardiomyocytes produced under different culture conditions at specific time points could make the manuscript more convincing.
  2. BM from AHCF is freeze-dried, which means that it may contain unknown growth factors or cytokines that promote cardiomyocyte differentiation. If the information related to BM analysis (e.g., preliminary analysis using cytokine array) can be provided, it will help others to understand more precisely the targets for further study in the future.

Minor concern
Some typos need to be fixed, e.g. NH4OH should have a subscript.

Author Response

Cardiac tissue-like 3D microenvironment enhances human fibroblast direct reprogramming into induced cardiomyocytes by microRNAs

Camilla Paoletti1,2,*, Elena Marcello1,2 , Maria Luna Melis1,2 , Carla Divieto3 , Daria Nurzynska4 and Valeria Chiono1,2

Response to Reviewers

We would like to express our gratitude to the Reviewers for their careful assessment, which allowed us to improve the quality of this paper. The Reviewers’ comments were carefully addressed: corresponding changes in the manuscript were implemented and highlighted in the revised version of the manuscript.

We have modified the manuscript’s title in: Cardiac tissue-like 3D microenvironment enhances route towards human fibroblast direct reprogramming into induced cardiomyocytes by microRNAs

Detailed answers to Reviewers’ comments are reported below.

Reviewer 2

This article describes the improvement of cardiomyocyte regeneration efficiency by microenvironmental strategies. The authors and their research team provide a detailed background and discussion of the experiments, and the steps and materials used to perform the experiments are clearly described. However, if some of the concerns could be further explained, it is believed that the article would be further enhanced.

Major concern:

  1. the authors only used cTnT staining to claim that it is induced cardiomyocyte, which we believe may not be strong enough. Since the authors' team has the ability to culture and confirm the physiological nature of cardiomyocytes (Paoletti et al., 2020), comparing the physiological and biochemical properties of cardiomyocytes produced under different culture conditions at specific time points could make the manuscript more convincing.

We thank the Reviewer for this comment. We added calcium transient studies in the paragraph 3.4, showing that miRcombo-treated cells cultured in 3D BM hydrogels exhibited increased percentage of cells showing Ca+2 transient, with increased slope of Ca+2 upstroke compared to miRcombo-treated cells in 3D hydrogels without BM.

  1. BM from AHCF is freeze-dried, which means that it may contain unknown growth factors or cytokines that promote cardiomyocyte differentiation. If the information related to BM analysis (e.g., preliminary analysis using cytokine array) can be provided, it will help others to understand more precisely the targets for further study in the future.

We thank the Reviewer for this comment. When we started to work with Biomatrix, we performed mass spectrometry analysis of total protein content (including cytokines and growth factors) contained in our cardiac-like matrix produced in vitro (data not shown). We observed the presence of Growth/differentiation factor 15 (GDF15), Tissue factor pathway inhibitor 2 (TFPI2) and Plasminogen activator inhibitor 1 (PAI-1), among other proteins such as Keratin, Vitronectin, Fibulin-2 (FBLN2) and Annexin 2 (ANXA2). We are aware that our immunofluorescence characterization, reported in figure 1C-D, is limited to just a small part of proteins that compose cardiac ECM. However, our aim was to show that BM production methods is an easy and low-cost procedure to obtain cardiac-like ECM in vitro, which retains main ECM proteins (fibronectin, laminin and collagens) after decellularization process. As we have shown that Biomatrix is critical for miRcombo-mediated reprogramming, together with 3D hydrogel culture, we may further investigate biomatrix composition in our next step and how its components affect miRcombo-mediated reprogramming.

Minor concern:

  1. Some typos need to be fixed, e.g. NH4OH should have a subscript.

We thank the Reviewer for this comment. The text has been thoroughly checked for typos and all the spelling mistakes have been corrected.

Round 2

Reviewer 1 Report

The author answered all my questions.

Author Response

Dear Reviewer,
we thank you for the comments. We have reported changes in figure 4b.